# SEDIQA: Sound Emitting Document Image Quality Assessment in a Reading Aid for the Visually Impaired

**DOI:** 10.3390/jimaging7090168

**Published:** 2021-08-30

**Authors:** Jane Courtney

**Affiliations:** School of Electrical & Electronic Engineering, Technological University Dublin, City Campus, Dublin, Ireland; jane.courtney@tudublin.ie

**Keywords:** image quality assessment, image quality metrics, NR-IQAs, D-IQA, OCR accuracy, OCR prediction, OCR improvements, visual aids, visually impaired, reading aids, document images, text-based images

## Abstract

For visually impaired people (VIPs), the ability to convert text to sound can mean a new level of independence or the simple joy of a good book. With significant advances in optical character recognition (OCR) in recent years, a number of reading aids are appearing on the market. These reading aids convert images captured by a camera to text which can then be read aloud. However, all of these reading aids suffer from a key issue—the user must be able to visually target the text and capture an image of sufficient quality for the OCR algorithm to function—no small task for VIPs. In this work, a sound-emitting document image quality assessment metric (SEDIQA) is proposed which allows the user to hear the quality of the text image and automatically captures the best image for OCR accuracy. This work also includes testing of OCR performance against image degradations, to identify the most significant contributors to accuracy reduction. The proposed no-reference image quality assessor (NR-IQA) is validated alongside established NR-IQAs and this work includes insights into the performance of these NR-IQAs on document images. SEDIQA is found to consistently select the best image for OCR accuracy. The full system includes a document image enhancement technique which introduces improvements in OCR accuracy with an average increase of 22% and a maximum increase of 68%.

## 1. Introduction

With advances in smartphone technology, particularly in camera quality, several visual aids for VIPs are emerging [1,2] with Microsoft’s Seeing AI as the current market front-runner. These assistive technologies range from navigation aids [3] to object detectors [4] and readers [5]. However, this last task has embedded in it reliance on OCR accuracy and, therefore, on image quality. This means that the user’s performance (hand motion, visual acuity, etc.) will affect the performance of the reader. Since these readers are both hand-held and designed for people with visual impairments, this is a fundamental issue that needs to be addressed.

To solve this issue, automatic processing can be done to improve OCR performance [6,7], but even the best performing pre-processors cannot achieve high OCR accuracy out of a low-quality image. Therefore, it is necessary to also assess the image quality before attempting OCR, and so a robust image quality assessment (IQA) metric is needed.

For this application, in the absence of a reference image, no-reference image quality assessment (NR-IQA)—otherwise known as “blind” IQA—is required. Most established NR-IQAs concentrate on perceptual image quality [8] but it has been found that these are not suitable for the application of document images, as the degradations that affect text-based content, and subsequently, OCR accuracy, can be quite different, and what is considered “high quality” in a scene image does not correlate with document image quality [9]. In fact, in previous work by the author [10], an unexpected reverse relationship was discovered between established NR-IQAs such as BRISQUE (Blind Referenceless Image Spatial Quality Evaluator [11]), NIQE (Natural Image Quality Evaluator [12]) and PIQE (Psychovisually-based Image Quality Evaluator [13]) and document image quality, with the metrics reporting lower quality results for ideal images than for their scanned counterparts.

A number of blind document image quality assessors (D-IQAs) have been developed to date. Some concentrate on specific degradations such as compression [14] or blur [15,16], while others concentrate on perceptual quality [17,18,19]. More recently, methods have tended towards learning [20,21,22,23]. However, trained networks can be slow and are affected by the size and diversity of the training dataset. Some promising work has emerged in the area of screen content quality [24,25,26] and, in this application area, a relationship was found between entropy and text-based image quality [27]. Other direct quality measures concentrate on the gradient of the image [28,29]. To date, no NR-IQA-based OCR accuracy predictors [30,31,32,33,34,35] pick up on the four major sources of OCR accuracy reduction: noise, blur, contrast and brightness. SEDIQA builds on the D-IQA findings, using entropy, gradient and median intensity to combine measures of the four main sources of error, creating a robust and directly measurable NR-IQA for documents.

As well as the metric, this work includes a document image enhancement technique with an emphasis on OCR accuracy improvements. Document image enhancement is still an open field of research and the SmartDoc competition [36,37] continues to encourage development in the area and to allow evaluation of document image enhancers and improvements on OCR accuracy. Some contributions have been made [16,38] using the associated dataset, which is also used here for comparison.

A systematic approach is taken to the investigation of OCR accuracy by first testing the relationship between accuracy and image degradations to determine which degradations should be the focus of the quality metric and the image enhancer. Performance of SEDIQA’s *Q*-metric was evaluated by comparing it against image degradations and OCR accuracy, as well as evaluating its performance alongside established NR-IQAs. The document image enhancer was evaluated by investigating improvements in OCR accuracy.

The full SEDIQA system is a visual reading aid design that automatically captures, assesses and converts camera-captured document images to audio outputs and ensures the best possible OCR accuracy for any given capture scenario.

The major contributions in this work include:(1)Testing of OCR accuracy versus image degradations to identify the degradations that contribute most significantly to OCR accuracy reduction.(2)A new, robust and directly measurable NR-IQA metric for document images. This is validated by testing on both synthetic and real images, against image degradations and OCR accuracy and alongside established NR-IQAs.(3)Insights into the performance of established NR-IQAs on document images.(4)Improvements in OCR accuracy in the full SEDIQA design.(5)SEDIQA as a visual reading aid.

## 2. Materials and Methods

This system was designed in Python 3.7 with OpenCV 4.3. Testing was done on a PC using Tesseract [39] for OCR and MATLAB 2020b for BRISQUE, NIQE and PIQE image quality tests. Tests were performed on synthetically degraded images, live captured images and the SmartDoc dataset [40] as an established benchmark. The SmartDoc dataset provides an excellent testbed for this design as it contains images of the same documents captured under different conditions and with different capture parameters—very much representing a realistic scanning scenario. The dataset contains images with both single and multiple distortions, which include variations in lighting, focus, motion blur, distance and perspective angle. These geometric and photometric distortions have significant negative effects on OCR accuracy. The dataset also includes text transcripts of the documents, allowing OCR accuracy to be measured.

The full dataset was tested but graphical results are presented for an individual document from this dataset (*D1*) for clarity. This document was selected as its scanned versions lead to a full range of OCR accuracies from 0 to 99%.

### 2.1. OCR Accuracy vs. Quality

OCR performance has been found to deteriorate significantly under real image degradations [41,42]. Some of these degradations are due to camera parameters and are constant for a given capture setup, others will vary with user performance (how accurately the user targets the document with the camera) and external conditions such as lighting. Compression and resolution belong to the former type and are constant, as the same smartphone camera is used to capture raw image data. However, noise, blur, contrast and brightness will change for each capture scenario. While contrast has been the emphasis of many document image binarization techniques [43], it has been shown to have relatively little effect on OCR performance, while noise and blur have been found to contribute most significantly to accuracy reduction.

To confirm this, testing is done on synthetic images with noise, blur, contrast reduction and brightness reduction introduced separately. These images are created initially in imaging software, with clear black text on a homogenous white background and represent ideal text images. Degradations are then introduced incrementally to study the effects of each type of degradation. A synthetic image and its associated histogram are shown in Figure 1 and samples of degraded versions of this are shown in Figure 2.

The set of synthetically degraded images was tested for OCR accuracy using the standard Characters Correct percentage as an accuracy measure. From the synthetic image tests, it is possible to ascertain which degradations have the most significant effect on OCR accuracy. These results were used in the development of the Q-metric and the Page Extractor, which is used to extract and enhance the text in the image.

### 2.2. SEDIQA Quality Measure

As can be seen in its image histogram (Figure 1b), an ideal text image is characterized by high median brightness (for dark text on a bright background), high contrast and low entropy. Entropy is visible as the spread of image values in the histogram, contrast approximates the width of the histogram and median brightness is the most common intensity value, in this case, white. The median intensity suffices for the majority of images and on the test dataset but will not work for bright text on dark backgrounds. To address this, a simple dominant intensity test can be used to determine whether to invert the image.

It has been shown that entropy, while considered proportional to quality in natural scene images, has a negative correlation with quality in text-based images, with higher entropy denoting lower quality [10,27]. As entropy captures both noise and blur deteriorations, it is potentially a good measure for predicting OCR performance:(1)$$E\left(I\right)=\overset{N}{\sum }p_{I}\log_{2}p_{I}$$

However, since entropy does not vary with brightness or contrast, brightness and contrast approximation measures are also needed. The median intensity, $\tilde{I}$, of the image is a good approximation of brightness and should be high in a good quality text-based image with a bright background. The standard deviation, *σ*, of the image approximates contrast, as it is proportional to the width of the histogram:(2)$$\sigma \left(I\right)=\sqrt{\frac{\sum {\left(I-\mu_{I}\right)}^{2}}{N}}\;\;$$

However, in real images, degradations are rarely constant throughout the image, so the entropy and standard deviation of the whole image are not useful. To overcome this issue, two versions of the image are acquired: the Entropy Image (EI) and the Gradient Image (GI).

Pixel values in the Entropy Image are local neighborhood average entropies for each location in the image. This local entropy should be high around text-content regions but low in the homogenous background. This means the median of the entropy image in a high-quality text image should be low while its standard deviation should be high. The Gradient Image captures the local contrast between the text and the background so in this image, again, the median should be low while the standard deviation should be high. In fact, the median of the Gradient Image is zero, so this term is omitted for computational efficiency.

Using these measures, SEDIQA’s quality metric is defined:(3)$$Q=\{\begin{array}{cc} \begin{array}{c} \frac{\tilde{I}+\sigma \left(EI\right)+\sigma \left(GI\right)}{\tilde{EI}} \end{array} & \;if\;\tilde{EI}>0\\ \tilde{I}+\sigma \left(EI\right)+\sigma \left(GI\right) & if\;\tilde{EI}=0 \end{array}$$
where $\tilde{I}=$ intensity median,$\;\tilde{EI}=$ entropy image median, σ(EI)= entropy image standard deviation and σ(GI)= gradient image standard deviation.

### 2.3. Validation of SEDIQA’s Q Metric

The *Q* values of the synthetically degraded images were measured to confirm the relationship between *Q* and image degradations noise, blur, contrast and brightness, as well as *Q*’s relationship with OCR accuracy. The *Q* values and OCR accuracies of the SmartDoc dataset were also measured to confirm the relationship between *Q* and OCR accuracy under real-world conditions. To compare *Q* with other measures, this same testing approach on both synthetic and real images was used with a set of well-established NR-IQAs: BRISQUE, NIQE and PIQE.

### 2.4. SEDIQA Design

The full SEDIQA design consists of three main stages: page extraction, quality measurement and audio output. As well as emitting a tone during capture to guide the user, the system automatically retains the highest quality image, allowing for automatic best-image selection. This image is then passed through an OCR algorithm, followed by text-to-speech software.

Until the page extraction stage is successful, the output will be a repeated ‘chirp’ and no image is retained. Once the page extractor successfully finds text, this is cropped, perspective warped and cleaned to create a page image, which is both saved as an initial best image and passed to the quality measurement stage. The quality of the image is measured, saved as an initial *Q*_MAX_ and converted to a tone where the frequency of the tone is proportional to the quality.

While SEDIQA is active, a new frame is grabbed from the camera and this process is repeated with an image added to the Best Image Array whenever its *Q* value exceeds *Q*_MAX_, at which point *Q*_MAX_ is updated to the minimum *Q* in the array.

The full workflow for the system can be seen in Figure 3.

#### 2.4.1. Page Extractor

A simple page extraction method is required to ensure that text has been found before a quality measure is taken. This method must be fast enough to ensure it can be implemented in near real-time but robust enough to operate under a variety of image degradations.

In this design, the image is first binarized using adaptive thresholding [44] followed by a dilation [45] to accentuate the boundary of the page. The largest contour in this image is taken as the page boundary and a polygon approximation is performed to establish the corners. While more sophisticated text detection methods, such as Google Vision or EAST [46], could replace this method, the design here was found to suffice for a typical page capture scenario, which is less challenging than the more general ‘Text in the Wild’ scenario and has been shown to work on the SmartDoc dataset and in live testing.

The extracted page boundary is then used in a perspective transform [47] to warp and crop the image to the page region only. This is then resized using a cubic interpolation and cleaned by contrast enhancing, sharpening and denoising.

The full workflow for the Page Extractor can be seen in Figure 4.

#### 2.4.2. Quality Measurement

In the quality measurement stage of SEDIQA, *Q* is measured once text has been found. For the first *N*_0_ images, the mean *Q* is calculated to act as a baseline, *Q*_0_. This is used for an approximate normalization of *Q* to ensure that the frequency falls into the audible range. It is also used as an initial maximum, *Q*_MAX_.
(4)$$Q_{N}=\frac{Q}{Q_{0}}$$

If *Q_N_* < *Q*_MAX_, no change occurs in *Q*_MAX_ and no image is captured; however, *Q_N_* is still calculated and passed to the audio output stage. If *Q_N_* > *Q*_MAX_, the image is retained as a member of the Best Image Array and *Q*_MAX_ is updated to the minimum *Q* in the array. The process is continued until enough images have been tested and the array is full.

#### 2.4.3. Audio Output

When the text is initially found by the Page Extractor, a neutral tone of 300 Hz is emitted. Once the baseline, *Q*_0_, is established (after the first *N*_0_ runs), *Q_N_* is calculated. Decreases in *Q_N_* are translated to lower frequency tones while increases are higher, tending towards the preferred 400–800 Hz range [48]. *Q_N_* is found to vary by about ±50% over a full range of OCR accuracies from 0 to 99%, so to stay in the audible range and close to the preferred frequency range, *Q**_N_* is scaled by 400 to convert it to Hz. The frequency of the tone in Hz is then given by:(5)$$f=400Q_{N}$$

The sound only acts as a guide for the user as the best images will be retained automatically. At the end of the capture process, the best image is passed to a Tesseract OCR algorithm and converted to text. If the confidence value returned by this algorithm is too weak, the “best image” can be rejected and the next best image used. When the confidence value is sufficiently high, the text extracted by this process is then passed to text-to-speech software to be read aloud.

### 2.5. Accuracy Improvements

To investigate the effect of the SEDIQA system on OCR accuracy, each document from the SmartDoc dataset was passed through the SEDIQA system and the OCR accuracy was measured on both the original images and the extracted page images. The full dataset was tested but for clarity, results are presented for a sample document (*D1*). This document was selected for presentation as it was found that the range of images of this document in the dataset, captured under different conditions, led to a full range of initial accuracies ranging from 0 to 99%. Although the full SEDIQA system only retains the best of these, the full range is presented to demonstrate the extent of the accuracy improvements introduced by the system.

## 3. Results

Before the full SEDIQA system was tested, the relationship between OCR accuracy and image degradations was investigated using synthetically created and degraded images of text (see Figure 1 and Figure 2). The *Q*-metric was validated on these synthetic images and its performance with respect to image degradations was compared with established NR-IQAs as well as with OCR accuracy. Note that in these images, *Q* does not need to be normalized for audibility, so the original *Q* value is used.

The OCR accuracy, *Q*-metric and established NR-IQAs were also tested on real camera-captured images from the SmartDoc dataset. The full SEDIQA system was tested on this dataset and in live capture to confirm the relationship between OCR accuracy and the *Q*-metric and to test the system’s accuracy improvements.

### 3.1. Synthetically Degraded Images

Using synthetically created text images, such as the one shown in Figure 1, the relationship between OCR accuracy and different forms of image degradation can be established. The level of degradation is increased from zero (original, ideal image) to a maximum and the OCR accuracy is tested for each level. Degradation is continued until OCR accuracy collapses, or a maximum degradation level is reached.

In previous work [41,42], it has been shown that noise and blur have significant effects on OCR accuracy while contrast and brightness have almost no effect. Although contrast and brightness show no effect on the OCR Accuracy in the ideal image case, variations in lighting throughout a real image can affect OCR performance and so these degradations are not ignored in developing the *Q*-metric.

The synthetic images were measured with the *Q*-metric and as a further validation, tested with well-established NR-IQAs: BRISQUE, NIQE and PIQE. Note that for these NR-IQAs, a lower value denotes higher quality. Results are presented here for each degradation type: noise, blur, contrast and brightness.

#### 3.1.1. Noise

To test the metric’s response to noise, Gaussian noise is incrementally added to the ideal image. The noise level is set by the sigma value, *σ*, in the probability density function:(6)$$G=\frac{1}{\sigma \sqrt{2\pi }}e^{-\frac{{\left(I-\mu_{I}\right)}^{2}}{2\sigma^{2}}}\;$$

The noise levels range from a value of 0.0125 for sigma at level 1 to 0.2375 at level 19, where OCR accuracy collapse occurs. Results are show in Figure 5. Although SEDIQA’s response to noise is a tad overdramatic, this ensures that images with high levels of noise would be rejected by the system and only the least noisy images would be retained as Best Image Array candidates. Of the other NR-IQAs, only PIQE shows the correct response to noise (decreasing quality with noise level). NIQE shows quality increasing with noise while BRISQUE shows almost no response (though it incorrectly shows lowest quality for the ideal image). Only SEDIQA and NIQE correctly select the ideal image as the best image.

#### 3.1.2. Blur

For blur, Gaussian blurring is used. This time, the blur level is set by the size of the kernel. As the blurring kernel must center on a pixel, it can only have odd values, so the number of data points is limited. The blur levels range from a kernel size of 3 × 3 (level 3) to 19 × 19 (level 19), though OCR accuracy collapse occurs around level 15.

Results are shown in Figure 6. Although *Q* erroneously shows an increase in quality at low blur, this is due to the median entropy in the ideal image being zero—a situation that does not arise in real images. In fact, the *Q*-metric tends towards infinity for an ideal image. BRISQUE and PIQE show the correct response to blur (decreasing quality with blur level), while NIQE shows quality increasing with blur.

#### 3.1.3. Contrast and Brightness

For contrast and brightness, the alpha and beta (gain and bias) parameters are used respectively:(7)D=αI+β 
where D= degraded image, I= original image, α= gain and β= bias.

For contrast, the gain is decreased until a difference of just one intensity level (in the range 0 to 255) between text and background is observed. The contrast levels range from an α of 0.2 for level 1 to 0.02 for level 19. For brightness, the bias is decreased until the intensity level is just 1 in this same range (0 to 255). The β range is from 0.95 for level 1 to 0.05 for level 19.

Results are shown in Figure 7 and Figure 8. The OCR Accuracy is not affected by either of these degradations in the synthetic image case, but it was found later in real image testing that images with lighting issues tended to perform poorly in OCR, and so these degradations were included in testing here. This is most likely due to non-uniform lighting across the document in real capture. SEDIQA’s response to both is linear, ensuring that images with lighting issues would be rejected by the system. Of the other NR-IQAs, only PIQE shows the correct response (decreasing quality with lighting issues). NIQE shows quality increasing with poor lighting, while BRISQUE shows almost no response.

These tests show SEDIQA’s *Q*-metric responding strongly to both noise and blur—the most significant factors in reducing OCR accuracy—whereas its responses to contrast and brightness are linear drop-offs. This means that the system will reject noisy, blurry and poorly lit images.

On an interesting side note, despite the fact that these established NR-IQAs continue to be used on text-based images, e.g., [49,50,51], it has been shown here that only PIQE responds correctly to these four common image degradations. This will be further investigated on real camera-captured images in the next section. For useful reference, a summary of these findings is presented in Table 1.

### 3.2. Real Camera-Captured Images

While the synthetic images allow individual degradations to be separated and examined, real camera-captured images are subject to random combinations of these degradations along with other distortions.

To test SEDIQA’s performance in real camera-captured images, the SmartDoc dataset was used, as it contains images of the same documents captured under different capture conditions. As it is a well-established benchmark, it also allows comparison with other metrics and previous work, e.g., [16,38].

To demonstrate SEDIQA’s performance, each version of the document was passed through the SEDIQA system. Although the system only retains a Best Image Array based on the *Q*-metric, the full set of results is presented here for completeness. This test was repeated for each document set as well as on live captured images.

Only one document set is presented in Figure 9 for clarity but a similar correlation between SEDIQA and OCR Accuracy is found throughout the dataset and across a variety of live capture scenarios.

For comparison, the images cleaned by the SEDIQA system were also tested using the established NR-IQAs. Results are presented in Figure 10. For ease of comparison, all metrics were normalized to the range 0 to 1.

Despite its under-performance in the image degradation tests, BRISQUE showed some correlation with OCR Accuracy, while PIQE and NIQE showed weak and positive correlations (again, for these metrics, negative correlation is correct). The comparative correlation results are shown in Table 2, with SEDIQA’s *Q*-metric showing the strongest correlation with OCR accuracy.

As the Best Image Array are the only candidates for text to speech conversion in the full SEDIQA system, their results are also presented in Table 3. Not only does the highest *Q* value correspond with the best performing image but all Best Image Array images give high accuracy.

Again, for comparison, the same Best Image Array selection method was performed using the other NR-IQAs and the OCR Accuracies of their top-ranking images were tested. Results are shown in Table 4 and confirm that, while BRISQUE shows some potential as an OCR Accuracy predictor, SEDIQA remains more robust and reliable.

### 3.3. Accuracy Improvements

As a final test of the full SEDIQA system, the OCR Accuracy of the original images was tested and compared to the accuracy of the cleaned images. The full set of results for document *D1* of the dataset are shown in Figure 11.

Although the system occasionally shows a decrease in accuracy, this is generally in images that are either of unusably poor quality or that do not pass the Page Extractor stage, where the image would be automatically rejected, and so these images are not passed to the Best Image Array.

The average increase in accuracy was 41% in the Best Image Array and 22% across the whole dataset, with a maximum increase for one image of 68% from just 25% accuracy in the original version to 94% in SEDIQA’s cleaned version, converting it from an unusable image to a candidate for the Best Image Array. The original image and cleaned version can be seen in Figure 12.

## 4. Discussion

Within this audio-based reading aid design lie some significant contributions to the field of Document Image Quality Assessment. First is a simple, robust and directly measurable NR-IQA for documents which picks up on four major sources of OCR accuracy reduction: noise, blur, contrast and brightness. This *Q*-metric has been validated by comparing to OCR accuracy, image degradations and three other established NR-IQAs: BRISQUE, PIQE and NIQE. The metric not only shows the strongest correlation with OCR accuracy, but also correctly selects the highest performing image as the best image from a large dataset of document scans of different qualities.

As part of the validation process, some interesting discoveries were also made about the established NR-IQAs. First, NIQE was shown to neither respond to typical image degradations nor correlate with OCR accuracy. This is not necessarily surprising as it is intended for natural images, yet it continues to be used in text-based images intended for OCR [50,52]. PIQE showed good responses to image degradations but in real document images had a reverse correlation with OCR accuracy, selecting some of the worst performing images. BRISQUE only responded to blur out of the degradations tested but did show some promise in real images where it showed some correlation with OCR accuracy. However, it was not robust and gave high quality scores to images which completely fail at conversion to text. Still, the combination of these findings about BRISQUE would suggest that blur may be the biggest contributor to OCR accuracy reduction.

As well as the robust metric, SEDIQA includes a text detection, extraction and cleaning process that leads to significant improvements in accuracy in even some of the poorest performing images. Across the entire SmartDoc dataset, the rejection rate at the page extraction stage was approximately 10% with almost 90% of images successfully cropped, warped and cleaned.

There are some minor limitations to be addressed in future work. This system has not yet been ported to a smartphone app and as such, has not yet been tested for speed, user experience (UX) or compatibility. However, these initial tests and the simplicity of the metric suggest that the design has significant potential.

SEDIQA could also be applied to camera focusing systems, text-in-the-wild applications (such as sign reading in autonomous cars or navigation aids) and any other text-based applications, particularly those involving OCR. As a Reading Aid, SEDIQA offers much needed audio guidance, to aid in document capture. Although other reading aids, such as Microsoft Seeing AI, offer some audio guidance in the text location stage, these do not assess, or feedback to the user, the quality of the image and as such, frequently lead to unsatisfactory results. As can be seen here, successful text detection does not necessarily mean successful OCR. With the addition of SEDIQA’s robust *Q*-metric to the reading aid design, the user can be sure of the best possible outcome from any given capture scenario.

## 5. Conclusions

SEDIQA automatically captures, assesses and converts camera-captured document images to audio outputs and ensures the best possible OCR accuracy from any given scan attempt. The major contributions in this work include:(1)Testing of OCR Accuracy vs. Image Degradations, identifying blur and noise as those that contribute most significantly to OCR accuracy reduction(2)A new, robust, directly measurable, validated NR-IQA for document images which consistently selects the best images for OCR accuracy performance(3)Insights into the performance of three well-established NR-IQAs on document images:
○BRISQUE was shown to only respond to blur out of the four major degradations tested, but performed reasonably well on real images. However, its failures were extreme, with accuracies less than 1% in its top ten images.○NIQE was found to be wholly unsuitable for document images.○PIQE responded to all four image degradations but completely failed on real images.(4)A document image enhancement technique leading to improvements in OCR accuracy of 22% on average across the whole SmartDoc dataset and a maximum increase of 68%.(5)The full SEDIQA Design as a Visual Reading Aid with audio outputs.

## Figures and Tables

**Figure 1 jimaging-07-00168-f001:**
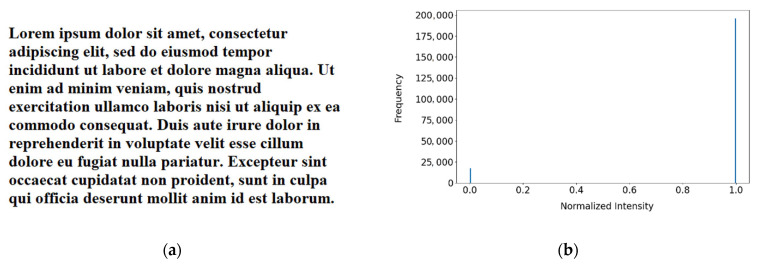
A sample ideal text image alongside its corresponding histogram: (**a**) ideal text image; (**b**) intensity histogram.

**Figure 2 jimaging-07-00168-f002:**
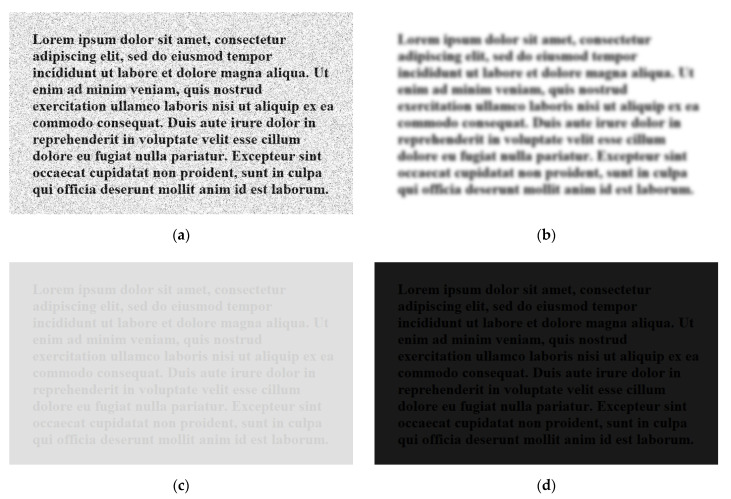
Degraded counterparts of the synthetic text image in Figure 1: (**a**) noisy (Gaussian noise added); (**b**) blurred (Gaussian blurring); (**c**) reduced contrast; (**d**) reduced brightness.

**Figure 3 jimaging-07-00168-f003:**
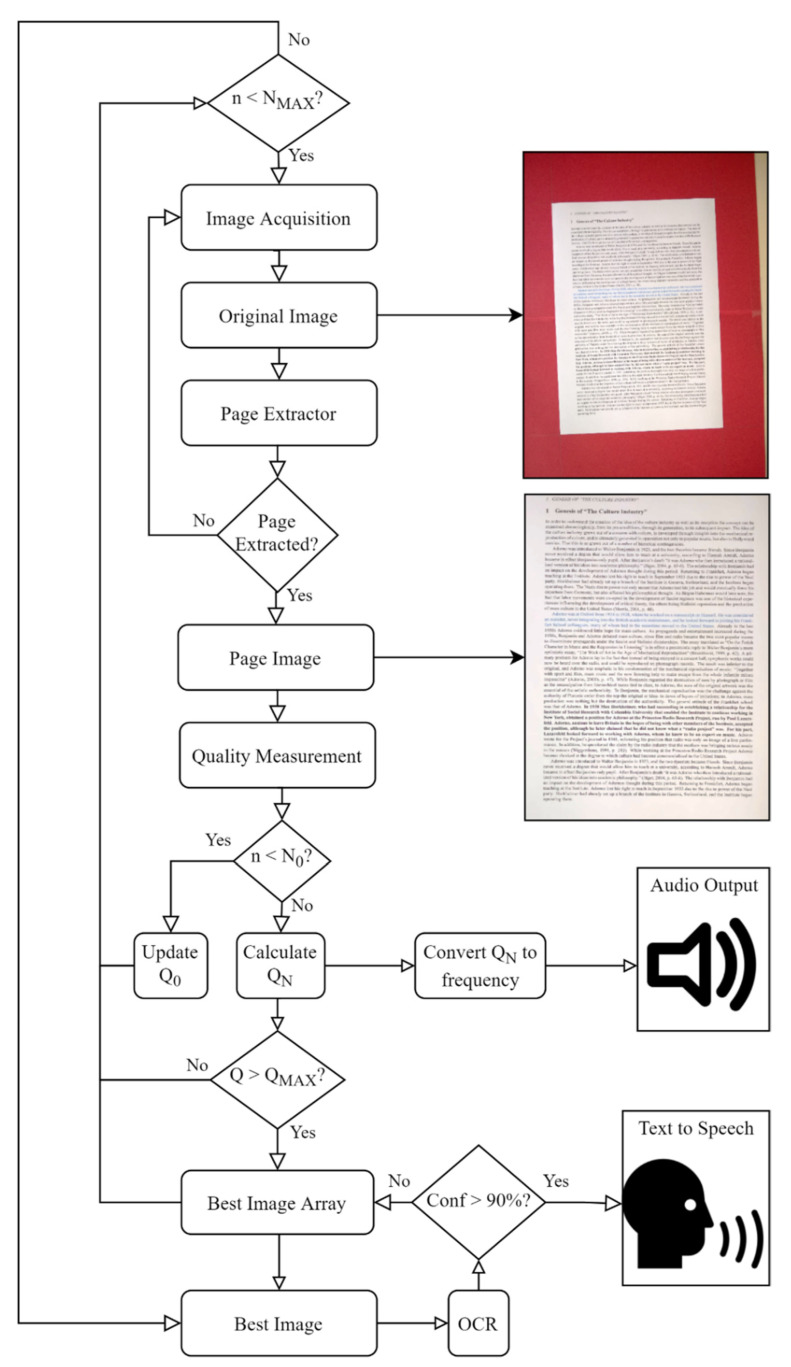
The workflow for the complete SEDIQA system; *n* = current image index, *N*_MAX_ = maximum no. of images to be captured, *N*_0_ = no. of baseline images to be captured, *Q*_0_ = baseline *Q*, *Q_N_* = normalized *Q*, *Q*_MAX_ = lowest *Q* in the Best Image Array, *Conf* = OCR confidence.

**Figure 4 jimaging-07-00168-f004:**
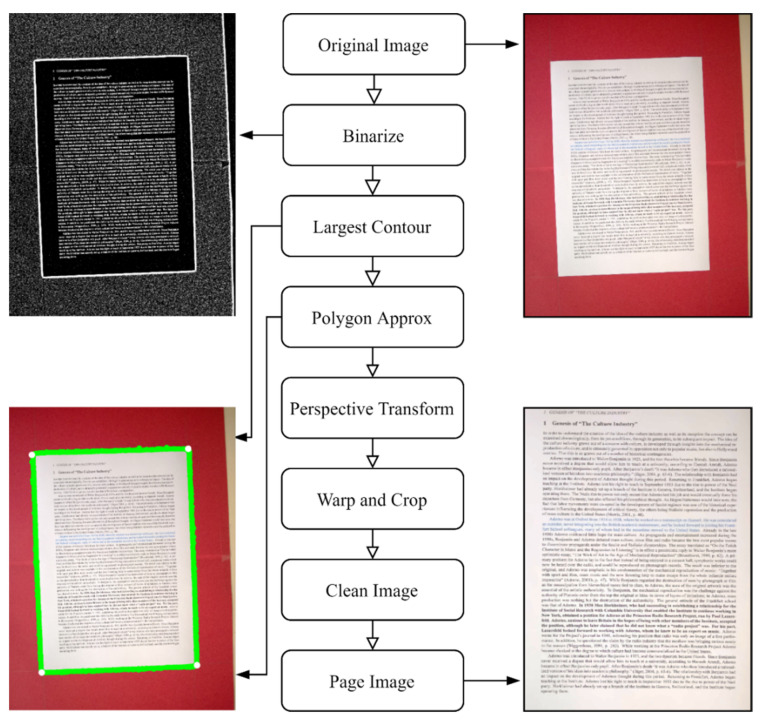
The workflow for the Page Extractor.

**Figure 5 jimaging-07-00168-f005:**
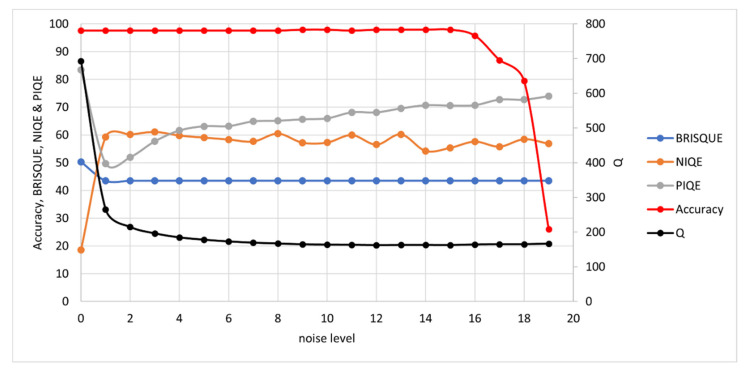
OCR Accuracy and Quality Metrics vs. Noise Level. Note: high BRISQUE, NIQE and PIQE values denote low quality while high *Q* values denote high quality. A separate axis is used for *Q* here as the unnormalized *Q* value tends to be considerably higher than the other NR-IQAs, which tend to stay in the range 0 to 100.

**Figure 6 jimaging-07-00168-f006:**
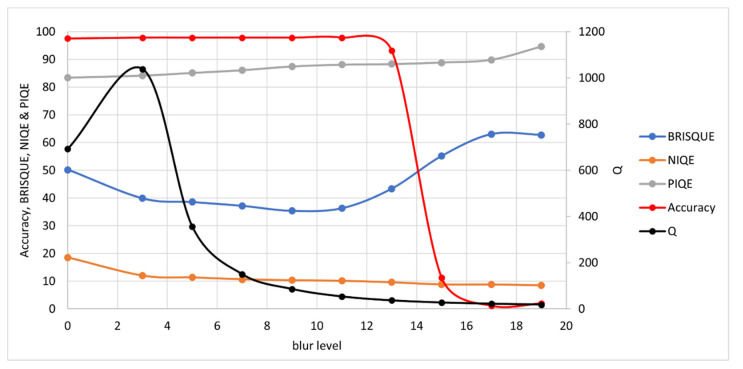
OCR Accuracy and Quality Metrics vs. Blur Level. Note: high BRISQUE, NIQE and PIQE values denote low quality while high *Q* values denote high quality. Again, a separate axis is used for *Q* here as the unnormalized *Q* value tends to be considerably higher than the other NR-IQAs, which tend to stay in the range 0 to 100.

**Figure 7 jimaging-07-00168-f007:**
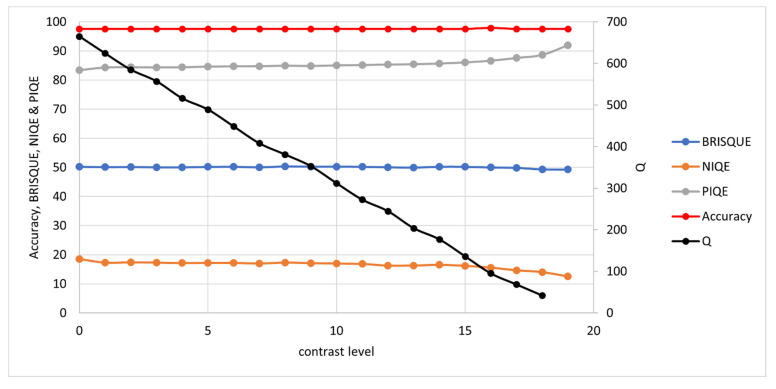
OCR Accuracy and Quality Metrics vs. Contrast Level. Note: high BRISQUE, NIQE and PIQE values denote low quality while high *Q* values denote high quality. Again, a separate axis is used for *Q* here as the unnormalized *Q* value tends to be considerably higher than the other NR-IQAs, which tend to stay in the range 0 to 100.

**Figure 8 jimaging-07-00168-f008:**
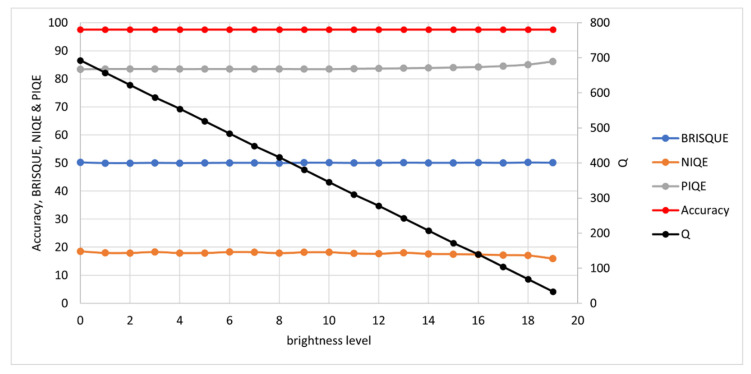
OCR Accuracy and Quality Metrics vs. Brightness Level. Note: high BRISQUE, NIQE and PIQE values denote low quality while high *Q* values denote high quality. Again, a separate axis is used for *Q* here as the unnormalized *Q* value tends to be considerably higher than the other NR-IQAs, which tend to stay in the range 0 to 100.

**Figure 9 jimaging-07-00168-f009:**
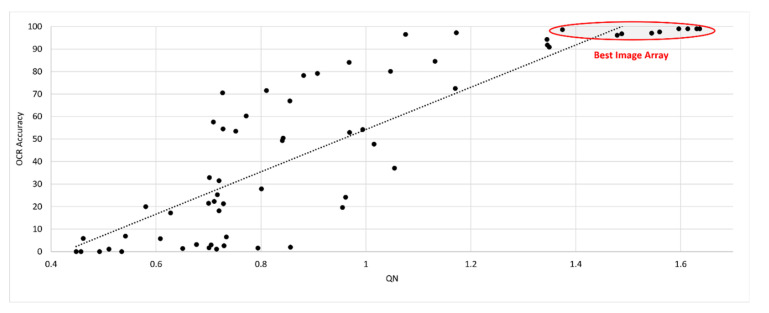
OCR Accuracy vs. *Q_N_* for document *D1* of the SmartDoc dataset. The system only retains those images in the Best Image Array for conversion to text, but the full results across all versions of the document are presented here.

**Figure 10 jimaging-07-00168-f010:**
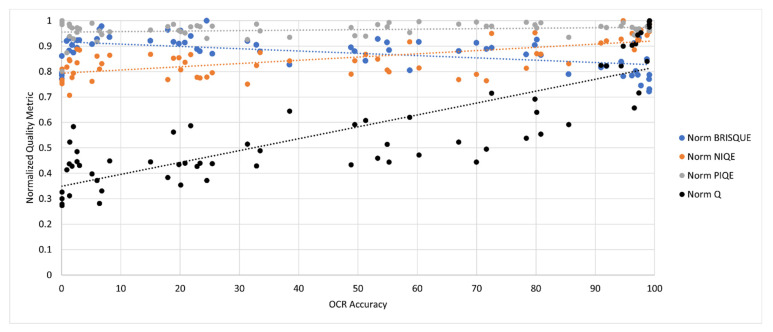
Normalized NR-IQAs vs. OCR Accuracy for document *D1* of the SmartDoc dataset.

**Figure 11 jimaging-07-00168-f011:**
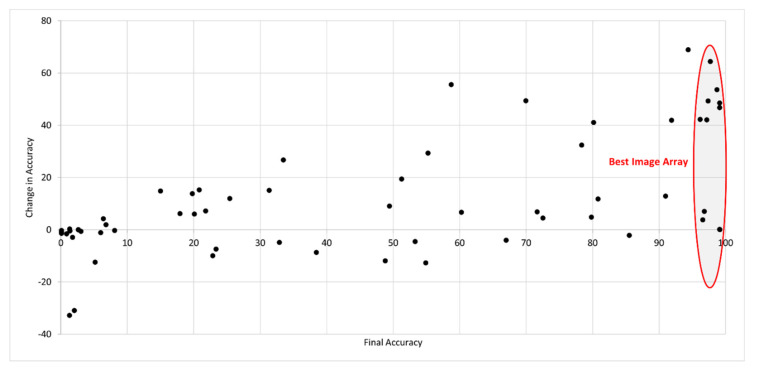
SEDIQA’s effect on OCR Accuracy for document *D1* of the SmartDoc dataset. Again, the system only retains those images in the Best Image Array for conversion to text but the full results across all versions of the document are presented here.

**Figure 12 jimaging-07-00168-f012:**
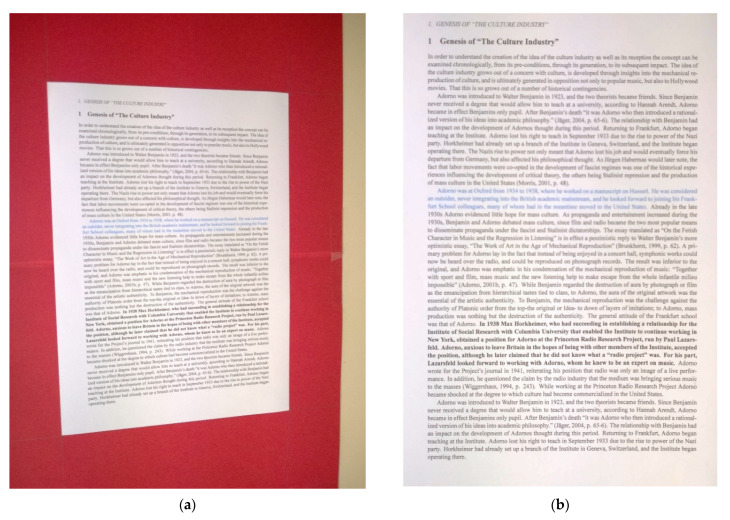
The most improved image of document *D1* of the SmartDoc dataset: (**a**) original version resulting in an OCR accuracy of 25%; (**b**) cleaned version with an OCR accuracy of 94%.

**Table 1 jimaging-07-00168-t001:** NR-IQAs and their ability to respond to image degradations in text-based images.

NR-IQA	Noise	Blur	Contrast	Brightness
BRISQUE	**🗴**	**🗸**	**🗴**	**🗴**
PIQE	**🗸**	**🗸**	**🗸**	**🗸**
NIQE	**🗴**	**🗴**	**🗴**	**🗴**
SEDIQA	**🗸**	**🗸**	**🗸**	**🗸**

**Table 2 jimaging-07-00168-t002:** Correlation of NR-IQAs with OCR Accuracy. Red denotes incorrect correlation.

NR-IQA	Correlation
BRISQUE	−0.5291
NIQE	0.6783
PIQE	0.2297
SEDIQA	**0.8463**

**Table 3 jimaging-07-00168-t003:** Normalized Q and OCR Accuracy results for the Best Image Array of document *D1*.

Image Rank	Accuracy	Norm Q
1	**99.11**	**1**
2	99.10	0.9967
3	99.08	0.9862
4	99.07	0.9757
5	97.68	0.9532
6	97.14	0.9440
7	96.80	0.9094
8	96.16	0.9039
9	94.69	0.8998
10	98.70	0.8402

**Table 4 jimaging-07-00168-t004:** Accuracies for best images of document *D1* as determined by each NR-IQA. Red denotes unusably low accuracies.

	Accuracies			
Image Rank	SEDIQA	BRISQUE	NIQE	PIQE
1	**99.11**	99.08	1.32	0.07
2	99.10	**99.11**	31.34	0.86
3	99.08	97.68	0.07	5.98
4	99.07	99.07	5.13	31.34
5	97.68	0.07	0.07	2.00
6	97.14	94.69	**71.63**	24.47
7	96.80	96.16	0.07	38.44
8	96.16	0.07	17.91	**85.50**
9	94.69	97.14	66.96	1.32
10	98.70	99.10	23.31	51.25

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
