# Peer review of "SEDIQA: Sound Emitting Document Image Quality Assessment in a Reading Aid for the Visually Impaired"

_2313-433X, 2021, doi:10.3390/jimaging7090168_

Round 1
Reviewer 1 Report
The authors have proposed a reading aid system for the visually impaired individuals. The proposed system SEDIQA is based on NR-IQA mechanism. The article is interesting and well written. A few improvements are recommended before considering for publication:
- There are some grammatical or language errors. The authors should carefully check throughout. For example, line 61 "This paper takes a systematic approach is taken to the investigation of OCR accuracy 61
by first testing the relationshi............" - The literature review should be added reviewing the latest NR-IQA based schemes. How the proposed technique differs from those, that would highlight the contributions more specifically.
- Conclusion section is missing. A short but clear conclusion should be added after discussion.
Author Response
The authors have proposed a reading aid system for the visually impaired individuals. The proposed system SEDIQA is based on NR-IQA mechanism. The article is interesting and well written. A few improvements are recommended before considering for publication:
Comment 1: There are some grammatical or language errors. The authors should carefully check throughout. For example, line 61 "This paper takes a systematic approach is taken to the investigation of OCR accuracy 61
by first testing the relationshi............"
Response: I hang my head in shame – the paper is indeed in need of some rigorous proofing! Mea culpa. I can only blame my rush to avoid our poor editors waiting any longer. My apologies for this, it should now be sorted.
Comment 2: The literature review should be added reviewing the latest NR-IQA based schemes. How the proposed technique differs from those, that would highlight the contributions more specifically.
Response: This comment also highlighted that some of my citations were missing, so many thanks for this. A paragraph has been added to the introduction, with the missing citations. See lines 57-61.
Comment 3: Conclusion section is missing. A short but clear conclusion should be added after discussion.
Response: A conclusion has been added, as requested. See Section 5.
Reviewer 2 Report
Turnitin report showing 20% plagiarism. Authors are requested to reduce the plagiarism of the paper as it is not acceptable for reputed journal.

Author Response
Turnitin report showing 20% plagiarism. Authors are requested to reduce the plagiarism of the paper as it is not acceptable for reputed journal.
Response: I do love a good Plagiarism Checker, but they do need to be watched. I was not surprised to find not a single word of plagiarism in this paper, since I typed every word with these very fingers. If the reviewer would be so kind as to open their own plagiarism report, they will find that the 20% “plagiarism” refers to the citations (which are indeed cited elsewhere), the journal template and the preprint of this very article!
Reviewer 3 Report
The paper introduces a quality assessment method for OCR approach to improve the quality of extracted texts.
I have some minor comments as following;
- BRISQUE , NIQE and PIQE are should be fully mentioned where they first appear.
* The SmartDoc dataset has to be detailed.
* I recommend including more figures which show the pros and cons of the proposed method, especially from the extracted pages and texts
Author Response
The paper introduces a quality assessment method for OCR approach to improve the quality of extracted texts.
I have some minor comments as following;
Comment 1: BRISQUE , NIQE and PIQE are should be fully mentioned where they first appear.
Response: This is very reasonable and has been done. See lines 45-47.
Comment 2: The SmartDoc dataset has to be detailed.
Response: Again, a reasonable request. This was originally done to some extent in the introduction to Section 2 but has been elaborated for clarity. See lines 93-97.
Comment 3: I recommend including more figures which show the pros and cons of the proposed method, especially from the extracted pages and texts.
Response: This is indeed a good idea. A sample has been added where the effects were significant and can be seen clearly. See Figure 12.
Reviewer 4 Report
The topic seems interesting, I have the following concerns to enhance the quality of the work.
- Authors should revise the abstract and accuracy should be added at end of the abstract.
- Why no reference image quality assessors give the best performance on documented images?
- The research problem/ requirement is not elaborated properly.
- Is the proposed approach is valid for colored document images??
- Authors need to re-write the Abstract in a more meaningful way example (Problem definition=> How existing methods are lacking => proposed solution => Outcome
- All equations should be assigned numbers. And align with the text.
- All figures should be redrawn with high resolutions and different colors.
- Authors should give all experiment parameters. Experimental setup, still few experiments paraments are missing??
Conclusion and Future work must be updated.
Author Response
The topic seems interesting, I have the following concerns to enhance the quality of the work.
Comment 1: Authors should revise the abstract and accuracy should be added at end of the abstract.
Response: Accuracy has been added to the Abstract, as requested.
Comment 2: Why no reference image quality assessors give the best performance on documented images?
Response: They don’t. They are simply necessary in the absence of a reference image, as stated in the Introduction: “For this application, in the absence of a reference image, No-Reference Image Quality Assessment (NR-IQA) – otherwise known as “blind” IQA – is required”.
Comment 3: The research problem/requirement is not elaborated properly.
Response: The research problem has been fairly thoroughly outlined so without specifics here, it is difficult to address this comment.
Comment 4: Is the proposed approach is valid for colored document images??
Response: Yes. As seen in the results.
Comment 5: Authors need to re-write the Abstract in a more meaningful way example (Problem definition=> How existing methods are lacking => proposed solution => Outcome
Response: The Outcome has been clarified in the Abstract, which now addresses these requirements as follows:
Problem definition: “For Visually impaired People (VIPs), the ability to convert text to sound can mean a new level of independence or the simple joy of a good book. “
How existing methods are lacking: “all of these reading aids suffer from a key issue – the user must be able to visually target the text and capture an image of sufficient quality for the OCR algorithm to function”
proposed solution: “a Sound-Emitting Document Image Quality Assessment metric (SEDIQA) is proposed which allows the user to hear the quality of the text image and automatically captures the best image for OCR accuracy”
Outcome: “SEDIQA is found to consistently select the best image for OCR accuracy. The full system includes a document image enhancement technique which introduces improvements in OCR accuracy with an average increase of 22% and a maximum increase of 68%.”
Comment 6: All equations should be assigned numbers. And align with the text.
Response: This is done in line with the journal template.
Comment 7: All figures should be redrawn with high resolutions and different colors.
Response: The requirements of the journal are a minimum 300 dpi and 1 MPixel resolution. All figures in this paper are twice the required dpi (600) and over 18 times the required resolution (the smallest is 18 MPixels).
Comment 8: Authors should give all experiment parameters. Experimental setup, still few experiments paraments are missing??
Response: The reported results are from the SmartDoc dataset to allow ease of comparison with other work. The details of this experimental set up have been elaborated in the introduction to Section 2.
Comment 9: Conclusion and Future work must be updated.
Response: A conclusion has been added. See Section 5.
Round 2
Reviewer 1 Report
The author has addressed most of the comments. The article can be accepted for publication. Proofread the article for any spelling or language errors.
Reviewer 2 Report
Accepted. Some formatting is required.